# Effect of ethnicity and other sociodemographic factors on attendance at diabetic eye screening: a 12-month retrospective cohort study

Abraham Olvera-Barrios [1,2] Michael Seltene,[1] Tjebo F C Heeren [1,2] Ryan Chambers [3] Louis Bolter [3] Adnan Tufail [1,2] Christopher G Owen [4] Alicja R Rudnicka [4] Catherine Egan [1,2] John Anderson [3]

¹Medical Retina, Moorfields Eye Hospital NHS Foundation Trust, London, UK
²University College London Institute of Ophthalmology, London, UK
³Homerton University Hospital NHS Foundation Trust, London, UK
⁴Population Health Research Institute, St George's, University London, London, UK

**Correspondence to**
Dr John Anderson;
john.anderson@nhs.net

## ABSTRACT

**Objectives** To examine the association of sociodemographic characteristics with attendance at diabetic eye screening in a large ethnically diverse urban population.

**Design** Retrospective cohort study.

**Setting** Screening visits in the North East London Diabetic Eye Screening Programme (NELDESP).

**Participants** 84 449 people with diabetes aged 12 years or older registered in the NELDESP and scheduled for screening between 1 April 2017 and 31 March 2018.

**Main outcome measure** Attendance at diabetic eye screening appointments.

**Results** The mean age of people with diabetes was 60 years (SD 14.2 years), 53.4% were men, 41% South Asian, 29% White British and 17% Black; 83.4% attended screening. Black people with diabetes had similar levels of attendance compared with White British people. However, South Asian, Chinese and 'Any other Asian' background ethnicities showed greater odds of attendance compared with White British. When compared with their respective reference group, high levels of deprivation, younger age, longer duration of diabetes and worse visual acuity, were all associated with non-attendance. There was a higher likelihood of attendance per quintile improvement in deprivation (OR, 1.06; 95% CI, 1.03 to 1.08), with increasing age (OR per decade, 1.17; 95% CI, 1.15 to 1.19), with better visual acuity (OR per Bailey-Lovie chart line 1.12; 95% CI, 1.11 to 1.14) and with longer time of NELDESP registration (OR per year, 1.02; 95% CI, 1.01 to 1.03).

**Conclusion** Ethnic differences in diabetic eye screening uptake, though small, are evident. Despite preconceptions, a higher likelihood of screening attendance was observed among Asian ethnic groups when compared with the White ethnic group. Poorer socioeconomic profile was associated with higher likelihood of non-attendance for screening. Further work is needed to understand how to target individuals at risk of non-attendance and reduce inequalities.

## Strengths and limitations of this study

► Strengths are that it is the largest population-based study in which the effect of sociodemographic factors on attendance at diabetic eye screening has been examined.

► A large urban multi-ethnic population with a broad range in its indices of deprivation was studied, making the findings widely applicable.

► Data completeness was high in the population.

► A range of additional socioeconomic factors, including the distance to screening centre and public transport accessibility level, were analysed.

► Systemic risk factors for diabetic retinopathy incidence and progression, and the association of sociodemographic variables with diabetic retinopathy were not available to analyse.

## INTRODUCTION

Diabetic retinopathy is a common neurovascular complication of diabetes and a major cause of blindness.[1 2] There are at least 3.9 million people diagnosed with diabetes in the UK, a number expected to rise to 5.8 million by 2025.[3] It is estimated that 30% of people with diabetes will develop retinopathy, and about 9% will develop sight-threatening retinopathy.[4] An early diagnosis through population screening, timely referral and treatment are essential for prevention of diabetes-related visual impairment.[5 6 7] The UK implemented the first systematically organised diabetic eye screening programme (DESP) in the world in England in 2003, achieving nationwide coverage by 2008. The English DESP offers annual mydriatic photographic screening to all people with diabetes aged 12 or older.[5] In accordance with national standards, screening of ≥85% of the eligible diabetic population is considered achievable, however, English DESP uptake data from 2016 to 2017 showed that this was not met in 75% of London's Clinical Commissioning Groups (CCGs) areas.[8 9] Regional differences

in screening delivery and uptake may explain regional variation in diabetic eye disease.[4]

Non-attendance at annual diabetic eye screening visits has been associated with late presentation of sight-threatening retinopathy.[7 10] Inequalities in health tend to be present in urban areas with contrasting sociodemographic conditions. London, a metropolis where people from the extremes of the deprivation indices live side-by-side, is a remarkable example of how these inequalities can result in different uptake rates across and within boroughs.[11–13] Health inequalities can create significant attendance variation among subgroups, and are of concern to any screening programme. Sociodemographic factors such as, age,[14–21] gender,[21–23] ethnicity,[14 15 23] transportation[24] and socioeconomic deprivation[14–16 19–21 23 25 26] have all been associated with non-attendance.

The North East London population is sociodemographically diverse, with a wide variation in ethnicities and a varied health profile with higher than average levels of deprivation and a lower than average life expectancy.[27–29] The North East London DESP (NELDESP) serves a total eligible population of approximately 125 000 people with diabetes aged 12 and over.[27] The NELDESP aims to invite ≥98% of eligible individuals and to have an uptake ≥85%. We have examined attendance at diabetic eye screening to identify which sociodemographic factors are determinants of attendance in this large multi-ethnic population with high levels of deprivation.

## METHODS

This was a 12-month retrospective cohort study of diabetic eye screening appointments occuring between 1 April 2017 to 31 March 2018. Our outcome measure was attendance at diabetic eye screening. Potential determinants of attendance include age, gender, self-defined ethnicity, area level deprivation, type of diabetes, duration of diabetes, visual acuity, years of NELDESP registration, distance to screening centre, and public transport accessibility.

### Setting

The NELDESP is provided by the Homerton University Hospital NHS Foundation Trust. We analysed data from people with diabetes living in six CCG areas with inner city multi-ethnic populations. These are the London boroughs of Newham, Redbridge, Tower Hamlets and Waltham Forest, classified as the most ethnically diverse in London[30 31]; the borough of Hackney and the borough Barking and Dagenham, which both have a substantial multi-ethnic population.

The NELDESP is run according to English DESP guidelines. All people with diabetes aged 12 or older are identified through coding in primary care electronic record systems. Over the course of one year, every person eligible for Routine Digital Screening is offered multiple opportunities to attend. Eligibility for screening is defined by Public Health England.[32] Software is used to generate invitations to attend for NELDESP appointments. The Homerton Hospital carries out appointment call/recall, screening, image grading, referral tasks, and is responsible for providing clinical leadership and programme management, including failsafe procedures and internal quality assurance.[27]

Briefly, a screening visit entails a visual acuity assessment, and pupil dilation to obtain two 45° digital retinal images of each eye, centred on the fovea and disc, respectively. We have described in detail the imaging, grading protocol and referral pathway elsewhere.[33]

Any person attending any of the offered appointments over the course of one whole year was defined as 'Attended'. Only those who failed to attend all appointments offered in the period were classified as 'Did not attend'.

### Data extraction

We carried out an anonymised data extraction of all screening appointments within the study period using structure query language (SQL) searches. An anonymised data base for analysis was created.

### Independent variable recording
#### Ethnicity

Self-classified ethnicity data were collected from patients at the time of screening, or from the routinely recorded ethnicity data provided by their general practitioner (GP) surgery. Their ethnicity was recorded in the nationally mandated screening software in accordance with the 2011 Office for National Statistics census groups.[30]

#### Index of multiple deprivation

The English indices of deprivation are composed of 39 indicators arranged in seven different domains of deprivation, which are combined and weighted to create the index of multiple deprivation (IMD), the official measure of relative deprivation in England. This measure is calculated for every neighbourhood or small area (lower-layer super output area (LSOA)) in England. There are 32 844 LSOAs with an average population of 1500, and each of them is ranked from 1st, the most deprived area, to 32, 844th, the least deprived area. Patients' postcodes were linked to their LSOA indices of multiple deprivation scores.

#### Visual acuity, distance and Public Transport Accessibility Level

We recorded the most recent visual acuity within a 3-year time frame in Snellen notation for the analysis. The better-seeing eye visual acuity score was assigned to each person. We calculated distance to screening centre (in kilometres) as a straight line from the patient's postcode to the screening site. For patients who attended, the postcode used was that known to the NELDESP on the day of attendance. For patients who failed to attend at any point within the study period, the postcode used was that known to the NELDESP on the date of the last offered appointment. The public transport accessibility level (PTAL) is a metric tool from Transport for London which rates locations by distance to the public transport network, thus reflecting the accessibility to public transport within

Greater London. The PTAL grade takes into account walk access time, average waiting time, service availability and service reliability. The grading has nine levels from 0 (with the poorest access) to 6b (excellent access).[34] Using Transport for London's Web-based Connectivity Assessment Toolkit (WebCAT),[35] we extracted the PTALs for each patient's home postcode.

## Statistical analysis

We used R V.4.0.0 for statistical analysis.[36] We conducted a multivariable logistic regression analysis of attendance at screening visit (binary outcome coded '1' if patient attended and '0' if they did not attend). A test for linear trend was performed if the ORs showed a linear pattern across categorical variables. Attendance was defined as a participant completing the diabetic retinopathy screening process. Independent variables considered were age, gender, ethnicity, IMD, type of diabetes, duration of diabetes, visual acuity, years of registration into the DESP, distance to screening centre and PTAL.

We categorised continuous variables for the analysis to allow for non-linear patterns in attendance. Rank scores of the IMD were split into quintiles following Office for National Statistics data of the English indices of deprivation 2019, with the 1st quintile being the most deprived and the 5th quintile the least deprived areas.[28] PTAL was divided into tertiles, with the 1st tertile having the worst PTAL (0, 1a, 1b) and 3rd tertile the best (5, 6a, 6b). Ethnicity was categorised as White British (White British, Irish, Any other White background), Mixed (White and Black Caribbean, White and Black African, White Asian, Any other Mixed background), Black (African, Caribbean, Any other Black background), South Asian (Indian, Pakistani, Bangladeshi), Chinese, Any other Asian background and Any other Ethnic group. Missing data points were categorised as 'Unknown' group within each independent variable.

The reference category for ethnicity was the White British group, for IMD the most deprived quintile, and for PTAL the best tertile. For the rest of the independent variables, the group with the highest number of observations was considered the reference.

## Patient and public involvement

Two patients provided insight into our discussion of the results of this study. We plan to disseminate the findings of our study to people eligible for diabetic eye screening and their families through the local press and via social media. In addition, we intend to seek wider dissemination to the public through the English national screening programme's communication team.

## RESULTS

A total of 84 449 people were invited for a screening appointment during the study period. Mean age was 60 years (SD 14.2 years), 53.4% were men and 93.7% of those invited for screening had type 2 diabetes. The majority were of South Asian ethnicity (41.2%), followed by White British (29%) and Black ethnic groups (17%). 74.7% of the participants lived in areas with the highest levels of deprivation (1st and 2nd IMD quintiles). Overall, screening attendance during the study period was 83.4%.

Table 1 summarises sociodemographic characteristics of attenders and non-attenders along with crude and adjusted ORs for attendance versus non-attendance (where ORs greater than 1 imply greater odds of attendance). In the text we refer to the adjusted OR from the multivariable linear regression model unless otherwise stated.

Those aged 12–45 years of age showed poorer attendance when compared with the reference 46–60 year-old group. In adjusted analyses, participants 18–30 years of age were least likely to attend for screening showing a 58% reduction in the odds of attendance, and an absolute uptake difference of 18.8% when compared with the reference. Each decade rise in age increased the odds of attendance by about 17% (OR= 1.17; 95%CI 1.15-1.19, p-value < 0.001).

Compared with White British individuals, those of Mixed or Black ethnicity did not show any difference in the odds of attendance after adjustment. However, odds of attendance were higher among individuals of Asian (South Asian, Chinese and any other Asian background) ethnicities when compared with White British individuals, even after adjustment.

Adjusted analyses showed that individuals living in the least deprived areas (5th IMD quintile) were most likely to attend for their screening appointments. Those in the 5th IMD quintile showed a 25% increase in the odds of attendance compared with people living in the most deprived areas (1st IMD quintile). Each IMD quintile increase (i.e. less deprivation) suggested a 6% rise in the odds of attendance (linear trend test p-value < 0.001).

People with longer duration of diabetes were less likely to attend. The OR per 5-year increase in duration of diabetes was 0.97 (95% CI 0.95 to 0.99, p value=0.019). The average distance to screening centre was 1.7 km (IQR 1–2 km). Only people who lived ≥9 km from the screening centre (outside the geographical boundaries of the CCGs of North East London) were less likely to attend. Odds of attendance decreased by 1% for every km further from the screening centre; a non-signficant trend (OR=0.99; 95% CI 0.97 to 1, p value=0.031).

Individuals with lower visual acuity (starting from visions worse than 6/9) showed a graded decline in the odds of attending the screening visit. Those with visual acuity worse than 6/18 were least likely to attend and showed a 60% reduction in odds of attendance compared with those with acuity of 6/6 to 6/9. This equates to an absolute difference in attendance of 11 percentage points when compared with the reference group.

Attendance did not appear to differ by gender, type of diabetes or PTAL score. People registered in the screening programme for more than 5 years were more likely to attend than those registered for less than 5 years.

**Table 1** Sociodemographic characteristics of attenders and non-attenders along with crude and adjusted odds ratios (OR) for attendance versus non-attendance

| Dependent: Attended* | Attended n=70405 (83.4%) | Did not attend n=14044 (16.6%) | Univariable OR (95% CI, p-value) | Multivariable adjusted OR† (95% CI, p-value) |
|---|---|---|---|---|
| **Age** | | | | |
| 12–17 years | 276 (78.9) | 74 (21.1) | 0.73 (0.57–0.95, **p=0.016**) | 0.71 (0.52–0.99, **p=0.036**) |
| 18–30 years†† | 1003 (64.9) | 543 (35.1) | 0.36 (0.32–0.40, **p<0.001**) | 0.42 (0.36–0.49, **p<0.001**) |
| 31–45 years | 8296 (77.2) | 2454 (22.8) | 0.66 (0.62–0.70, **p<0.001**) | 0.71 (0.66–0.76, **p<0.001**) |
| 46–60 years (Reference) | 25779 (83.7) | 5029 (16.3) | – | – |
| 61–75 years | 24482 (86.4) | 3856 (13.6) | 1.24 (1.18–1.30, **p<0.001**) | 1.28 (1.21–1.35, **p<0.001**) |
| 76–90 years | 10109 (84.0) | 1930 (16.0) | 1.02 (0.97–1.08, p=0.461) | 1.20 (1.11–1.29, **p<0.001**) |
| >90 years†† | 460 (74.4) | 158 (25.6) | 0.57 (0.47–0.68, **p<0.001**) | 0.92 (0.73–1.17, p=0.487) |
| Per decade (Mean (SD)) | 6.0 (1.4) | 5.8 (1.6) | 1.14 (1.13–1.15, **p<0.001**) | 1.17 (1.15–1.19, **p<0.001**) |
| **Gender** | | | | |
| Male (Reference) | 37569 (83.3) | 7558 (16.7) | – | – |
| Female | 32836 (83.5) | 6486 (16.5) | 1.02 (0.98–1.06, p=0.323) | 0.99 (0.95–1.04, p=0.717) |
| **Ethnicity** | | | | |
| White British (Reference) | 20040 (81.9) | 4435 (18.1) | – | – |
| Mixed | 845 (77.7) | 242 (22.3) | 0.77 (0.67–0.90, **p=0.001**) | 0.90 (0.75–1.09, p=0.264) |
| Black | 11869 (82.9) | 2454 (17.1) | 1.07 (1.01–1.13, **p=0.014**) | 1.02 (0.95–1.09, p=0.590) |
| South Asian | 29708 (85.4) | 5084 (14.6) | 1.29 (1.24–1.35, **p<0.001**) | 1.16 (1.09–1.23, **p<0.001**) |
| Chinese | 536 (89.8) | 61 (10.2) | 1.94 (1.50–2.56, **p<0.001**) | 1.91 (1.39–2.71, **p<0.001**) |
| Any other Asian background | 4683 (88.0) | 640 (12.0) | 1.62 (1.48–1.77, **p<0.001**) | 1.30 (1.17–1.45, **p<0.001**) |
| Any other ethnic group | 2248 (83.0) | 460 (17.0) | 1.08 (0.97–1.20, p=0.145) | 1.05 (0.92–1.20, p=0.453) |
| Unknown†† | 476 (41.6) | 668 (58.4) | 0.16 (0.14–0.18, **p<0.001**) | 0.32 (0.27–0.38, **p<0.001**) |
| **IMD quintiles** | | | | |
| 1st (most deprived, reference) | 20136 (81.9) | 4456 (18.1) | – | – |
| 2nd | 32163 (83.5) | 6359 (16.5) | 1.12 (1.07–1.17, **p<0.001**) | 1.09 (1.04–1.15, **p=0.001**) |
| 3rd | 12196 (84.7) | 2203 (15.3) | 1.23 (1.16–1.30, **p<0.001**) | 1.17 (1.10–1.26, **p<0.001**) |
| 4th | 4457 (85.1) | 778 (14.9) | 1.27 (1.17–1.38, **p<0.001**) | 1.15 (1.04–1.27, **p=0.009**) |
| 5th (Least deprived) | 1453 (85.4) | 248 (14.6) | 1.30 (1.13–1.49, **p<0.001**) | 1.25 (1.06–1.50, **p=0.012**) |
| Per quintile (Median (IQR)) | 2 (1–4) | 2 (1–3) | 1.08 (1.05–1.10, **p<0.001**) | 1.06 (1.03–1.08, **p<0.001**) |
| **Type of diabetes** | | | | |
| Type 1 DM | 2223 (75.8) | 710 (24.2) | 0.55 (0.51–0.60, **p<0.001**) | 1.09 (0.96–1.25, p=0.190) |
| Type 2 DM (Reference) | 67265 (85.0) | 11851 (15.0) | – | – |
| MODY | 40 (81.6) | 9 (18.4) | 0.78 (0.40–1.72, p=0.508) | 0.85 (0.40–2.07, p=0.687) |
| Not specified/other†† | 877 (37.3) | 1474 (62.7) | 0.10 (0.10–0.11, **p<0.001**) | 0.46 (0.40–0.53, **p<0.001**) |
| **Duration of diabetes** | | | | |
| 1–10 years (Reference) | 44890 (83.6) | 8778 (16.4) | – | – |
| 11–20 years | 20327 (86.3) | 3236 (13.7) | 1.23 (1.18–1.28, **p<0.001**) | 0.99 (0.92–1.06, p=0.727) |
| >20 years | 5057 (83.8) | 977 (16.2) | 1.01 (0.94–1.09, p=0.743) | 0.87 (0.78–0.97, **p=0.011**) |
| Unknown†† | 131 (11.1) | 1053 (88.9) | 0.02 (0.02–0.03, **p<0.001**) | 0.35 (0.26–0.47, **p<0.001**) |
| Per 5 years (Mean (SD)) | 1.9 (1.5) | 1.7 (1.5) | 1.05 (1.04–1.07, **p<0.001**) | 0.97 (0.95–1.00, **p=0.019**) |
| **Distance to centre** | | | | |
| ≤1–2 km (Reference) | 55436 (83.8) | 10752 (16.2) | – | – |
| 3–5 km | 12895 (82.4) | 2758 (17.6) | 0.91 (0.87–0.95, **p<0.001**) | 0.97 (0.91–1.03, p=0.301) |
| 6–8 km | 1044 (80.4) | 254 (19.6) | 0.80 (0.70–0.92, **p=0.001**) | 0.90 (0.75–1.09, p=0.268) |
| ≥9 km | 190 (75.7) | 61 (24.3) | 0.60 (0.46–0.81, **p=0.001**) | 0.66 (0.46–0.97, **p=0.027**) |
| Unknown | 840 (79.3) | 219 (20.7) | 0.74 (0.64–0.87, **p<0.001**) | 0.93 (0.77–1.12, p=0.433) |
| Per km (Mean (SD)) | 1.7 (1.6) | 1.8 (1.7) | 0.97 (0.96–0.98, **p<0.001**) | 0.99 (0.97–1.00, **p=0.031**) |

**Table 1** Continued

| Dependent: Attended* | Attended n=70405 (83.4%) | Did not attend n=14044 (16.6%) | Univariable OR (95% CI, p-value) | Multivariable adjusted OR† (95% CI, p-value) |
|---|---|---|---|---|
| **PTAL tertiles** | | | | |
| 1st tertile | 23281 (83.2) | 4714 (16.8) | 0.95 (0.90–1.01, p=0.083) | 0.95 (0.89–1.02, p=0.189) |
| 2nd tertile | 36535 (83.4) | 7291 (16.6) | 0.96 (0.91–1.02, p=0.192) | 0.97 (0.90–1.03, p=0.309) |
| 3rd tertile (Reference) | 10589 (83.9) | 2039 (16.1) | – | – |
| **Visual acuity** | | | | |
| Better than 6/6 | 14069 (88.7) | 1798 (11.3) | 0.93 (0.88–0.98, **p=0.007**) | 1.08 (1.02–1.15, **p=0.007**) |
| 6/6 to 6/9 (Reference) | 52035 (89.4) | 6158 (10.6) | – | – |
| <6/9 to 6/18 | 3459 (84.7) | 626 (15.3) | 0.65 (0.60–0.72, **p<0.001**) | 0.60 (0.55–0.66, **p<0.001**) |
| Worse than 6/18†† | 683 (78.4) | 188 (21.6) | 0.43 (0.37–0.51, **p<0.001**) | 0.40 (0.34–0.48, **p<0.001**) |
| Per five letters(Mean (SD)) | 16.5 (1.6) | 16.3 (2.0) | 1.07 (1.06–1.08, **p<0.001**) | 1.12 (1.11–1.14, **p<0.001**) |
| **Years of registration** | | | | |
| 1–5 years (Reference) | 28809 (80.9) | 6822 (19.1) | – | – |
| 6–10 years | 22948 (84.8) | 4103 (15.2) | 1.32 (1.27–1.38, **p<0.001**) | 1.13 (1.07–1.20, **p<0.001**) |
| 11–15 years | 18242 (85.6) | 3072 (14.4) | 1.41 (1.34–1.47, **p<0.001**) | 1.22 (1.12–1.33, **p<0.001**) |
| 16–20 years | 406 (89.6) | 47 (10.4) | 2.05 (1.53–2.81, **p<0.001**) | 1.94 (1.35–2.89, **p=0.001**) |
| Per year(Mean (SD)) | 6.9 (3.9) | 6.3 (3.9 | 1.04 (1.04–1.05, **p<0.001**) | 1.02 (1.01–1.03, **p<0.001**) |

Observations are for 84449 individuals.
Bold p values indicate statistically significant results
Independent variables with missing data categorised as "Unknown": type of diabetes (2.8%), duration of diabetes (1.4%), distance to screening centre (1.3%), and ethnicity (1.4%).
*ORs greater than one imply greater odds of attendance.
†ORs mutually adjusted for all factors shown in the table.
‡Variable groups with uptake below the national diabetic eye screening programme uptake goal of≥75%. Independent variables with missing data categorised as "Unknown": type of diabetes (2.8%), duration of diabetes (1.4%), distance to screening centre (1.3%), and ethnicity (1.4%).
IMD, Index of Multiple Deprivation; MODY, Maturity onset diabetes of the young; PTAL, Public Transport Accessibility Level.

People with >15 years of registration showed almost twice the odds of attendance than people with <5 years of registration. The OR per 5 years of registration was 1.02 (95% CI 1.01 to 1.03, p value<0.001).

## DISCUSSION

We found that people of Mixed or Black ethnicity with diabetes show very similar likelihoods of attendance at diabetic eye screening appointments compared with White British individuals. People of all asian ethnicities were more likely to attend than White British people in this large, well organised, sociodemographically diverse urban DESP. This is the most current study with large scale data on ethnicity and diabetic eye screening. In addition, those with poorer visual acuity, younger age and residing in areas with higher levels of deprivation were less likely to attend for diabetic eye screening appointments.

### Principal findings and comparison with other studies

Black, Asian and minority ethnic (BAME) groups have been reported to be more likely to develop diabetic retinopathy than White Europeans, more likely to present with sight-threatening retinopathy,[15 37 38] and less likely to attend for diabetic eye screening.[14 15 23] Attendance rates for BAME groups in our study were all higher than the White British, except for the small Mixed ethnic group, which had a lower, though non-significant, rate

of attendance (4.2% uptake difference). Chinese, South Asian and any other Asian background ethnicities were most likely to attend, more so than any other ethnic group. These findings suggest that the underlying increased rates of retinopathy and sight-threatening retinopathy reported in BAME ethnic groups[38 39] are not explained by non-attendance, raising the issue of increased susceptibility or poorer diabetic control. A study by Gulliford et al.[15] analysing sociodemographic inequalities in diabetic eye screening in South London also reported that African, Caribbean and other ethnicity groups were more likely to attend for diabetic eye screening than White Europeans. Of note is that there was a higher proportion of missing ethnicity data in their study when compared to ours (~39% vs 1.4%, respectively).

The commissioning and provision of diabetes eye screening programmes in England has improved since previous analyses were conducted, this may explain some of the difference between this and previous studies. The cultural and language barriers perceived to prevent older people from BAME groups attending in previous studies have proven to be misplaced. All appointment letters are written in English, these data show that the language of the letter was no barrier to better attendance. Indeed, uptake was higher among older people and those from BAME groups.

Socioeconomic deprivation has consistently been associated with attendance, where those from more deprived areas are less likely to attend for eye screening appointments.[14 16 19 20 25 26] Although the overall average difference in attendance of 3.5% between most and least deprived areas found in our study is less than the 9.3% reported in earlier studies,[25] this is still greater than the 2% uptake difference found in a population from South London in 2010.[15] Our results provide further evidence of the ingrained health inequalities present in a multiethnic study population with high levels of deprivation. Also, we show the effect of multiple risk factors that appear to impact on attendance. Longer duration of diabetes and worsening visual acuity showed an association with non-attendance compared with individuals with shorter disease duration and better visual acuity. Previous reports have shown an association of longer duration of diabetes with non-attendance.[15 16] Given that duration of diabetes is one of the three major risk factors for diabetic retinopathy,[4 40 41] and considering that >60% of people with type 2 diabetes and almost all people with type 1 diabetes will have diabetic retinopathy after 20+ years duration of the disease,[40] the reduced odds of attendance observed in this group places them at increased risk of visual complications. There is, to our knowledge, no evidence available about the association of visual acuity with attendance to diabetic eye screening.

In other areas of the UK, increased distance from screening clinic has been associated with an increased risk of non-attendance.[19 37] We have found that only individuals living ≥9 km from a screening centre were less likely to attend. It is noteworthy that an 8 km radius from one of the NELDESP screening centre covers all of the geographical areas of these six CCGs. People who are living more than 8 km from a screening centre have moved outside the CCG area and but remain registered with a North East London GP . Interestingly, we found that the association of distance to screening centre with non-attendance is independent from PTALs in this inner-city population. This may be due to London having a well-developed public transport network and good transport-related access. These findings may not apply elsewhere, particularly to non-urban populations less served by public transport.

In accordance with previous evidence,[10 14–16 19 20 42 43] young individuals from 12 to 45 years of age had lower odds of attendance compared with people aged 46–60 years. Possible underpinning factors are over confidence about their health or demanding work schedules.[19 24] Nonetheless, within the context of diabetes chronicity and the need for regular contact with healthcare services, these individuals are at increased risk of complications through longer duration of disease and possible suboptimal metabolic control.[44]

Our study has several strengths. First, a large sample size with considerable proportions of individuals from different ethnic groups representing a diverse population group all living within the programme area, with one of the most complete data sets on ethnicity reported to date. Second, the use of PTALs in addition to distance to screening centre to evidence the associations of accessibility and transport with attendance. And third, the fact that three-quarters of the participants were distributed between two of the most deprived quintiles of IMD, allowing the comparative association between deprivation and ethnicity with attendance to be examined. Our study has several limitations. First, major systemic risk factors for diabetic retinopathy incidence and progression, namely hypertension and glycaemic control, were not available to include in our analysis. Second, we did not analyse the association of the sociodemographic variables with the presence of diabetic retinopathy, which although desirable, would have been difficult to ascertain for repeated non-attenders.

Further work to unravel the interplay between ethnicity, deprivation and disease severity, is needed to inform strategies to improve attendance, particularly in high risk under privileged groups.

## CONCLUSION

Smaller previous studies have reported an association between non-White ethnicities and poor attendance at diabetic eye screening appointments. However, in this large diverse urban population, South Asian, Chinese and individuals of Any other Asian background were more likely to attend for diabetic eye screening than White British people. Public health strategies have in the past focused on ethnic differences as a possible cause of variance in diabetic eye screening uptake. The data from this large cohort shows that there are other more influential factors. We have shown that worse visual acuity, higher levels of deprivation, younger age and longer duration of diabetes are more predictive of non-attendance. Hence, strategies to improve uptake should be directed at these groups, in order to reduce inequalities in diabetic eye screening.

**Contributors** All authors meet the ICMJE criteria for authorship. JA, CE, LB, AT, ARR, CGO and AO-B: designed the study. AO-B, MS, TFCH and RC: undertook data management, processing and analysis. ARR, CGO and CE provided statistical advice and analysed the data. JA, CE, LB, ARR and AO-B wrote the first draft of the report, which was critically appraised by all authors. All the authors read and approved the final draft for journal publication. JA is responsible for data integrity.

**Funding** This research has received a proportion of its funding from the Department of Health's NIHR Biomedical Research Centre for Ophthalmology at Moorfields Eye Hospital and UCL Institute of Ophthalmology (salary support for A.T. and C.E.), and from the Mexican National Council of Science and Technology (CONACYT, scholarship #2018-000009-01EXTF-00573 to AO-B).

**Disclaimer** The views expressed in the publication are those of the authors and not necessarily those of the Department of Health.

**Competing interests** None declared.

**Patient and public involvement** Patients and/or the public were involved in the design, or conduct, or reporting, or dissemination plans of this research. Refer to the Methods section for further details.

**Patient consent for publication** Not required.

**Ethics approval** This study was registered as an audit and approved through the research governance process at the Homerton University Hospital NHS Foundation Trust and adhered to the UK Data Protection Act 2018.

**Provenance and peer review** Not commissioned; externally peer reviewed.

**Data availability statement** Data are available upon reasonable request. The data that supports the findings of this study are available from the North East London Diabetic Eye Screening Programme upon reasonable request.

**ORCID iDs**
Abraham Olvera-Barrios http://orcid.org/0000-0002-3305-4465
Tjebo F C Heeren http://orcid.org/0000-0001-5297-2301
Ryan Chambers http://orcid.org/0000-0001-5322-4209
Louis Bolter http://orcid.org/0000-0002-8710-8916
Adnan Tufail http://orcid.org/0000-0001-6131-7640
Christopher G Owen http://orcid.org/0000-0003-1135-5977
Alicja R Rudnicka http://orcid.org/0000-0003-0369-8574
Catherine Egan http://orcid.org/0000-0002-4439-3489
John Anderson http://orcid.org/0000-0002-2355-9742

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
