## [Reviewer comments · BMJ Open]

ARTICLE DETAILS

TITLE (PROVISIONAL)	The effect of ethnicity and other sociodemographic factors on attendance at diabetic eye screening: a 12-month retrospective cohort study
AUTHORS	Olvera-Barrios, Abraham; Seltene, Michael; Heeren, Tjebo; Chambers, Ryan; Bolter, Louis; Tufail, Adnan; Owen, C; Rudnicka, Alicja; Egan, Catherine; Anderson, John

VERSION 1 – REVIEW

REVIEWER	Muhamad, Nor Asiah Institute for Public Health
REVIEW RETURNED	09-Feb-2021

GENERAL COMMENTS	Dear Authors, Congratulation on your write up of this manuscript. Below are my comments: 1. The objectives of your study are unclear. I assumed that you want to determine the association between sociodemographic and other related factors and attendance at annual diabetic eye screening. Please clearly state your objectives.2. You stated that your study design is retrospective cohort. However, I don't see any follow-up for patients attending the screening clinic. Can you explain further on the study variables/follow-up?3. The method is unclear. If it is a cohort study, I assume the follow-up the annual screening for each respondents. What did you do when patients come for repeated screening. Can you elaborate further on this?4. If this is a cohort study, it should be relative risk not odd ratio with different time point.
--

REVIEWER	Hasan, Md. Mehedi University of Queensland, Institute for Social Science Research
REVIEW RETURNED	28-Feb-2021

GENERAL COMMENTS	This article is generally well-constructed and well-written. The findings and interpretations are interesting and seem to be grounded in the data. The results drawn from the data seems well-linked with the recommendations that may have role in further reducing health inequality on the topic. However, the authors may consider some minor issues as given below: 1. "people with diabetes" may be replaced by "patients with diabetes" or "diabetic patients" throughout the paper2. There should not be space between estimate and % in page 2 line 26
---

	3. There should be a space between 95% and CI in page 2 line 34-36 4. Outcome variable should be clearly stated in the text, should be in a separate paragraph. 5. Is there any cases for whom the residence was changed and result the change in their distance to screening center? If yes, how the distance was then adjusted? 6. If possible, some variables such as medication intake for diabetes, controlled/uncontrolled diabetes should have an impact on attendance for screening, and hence should be considered 7. In page 6 line 24, which test was performed to examine trend of odds whether it is linear or not should be clearly mentioned 8. In page 7 line 36, not sure how the 6% rise was calculated. Would it be better to include a bit detail of the calculation? 9. The results of regression analysis were interpreted for univariable model in some places (for example, interpretation for visual acuity in page 7, line 53) and for multivariable model in other places (for example, interpretation for age in page 7, line 22). Should it be consistent? 10. In the table, the results of regression analysis were presented when the independent variables were considered as factors, whereas some of these variables were not considered as factor variables when they were interpreted (e.g. age, distance etc). Should it be consistent with the estimates presented and those interpreted in the text? 11. Some references are too old to use (such as ref 5, ref 6). Better to use updated one if possible.
--	--

VERSION 1 – AUTHOR RESPONSE

Reviewer: 1

Dr. Nor Asiah Muhamad, Institute for Public Health

Comments to the Author:

Dear Authors,

Congratulation on your write up of this manuscript.

- Thank you

Below are my comments:

1. The objectives of your study are unclear. I assumed that you want to determine the association between sociodemographic and other related factors and attendance at annual diabetic eye screening. Please clearly state your objectives. Thank you, we have clarified this objective both in the abstract and at the end of our introduction section as recommended.

Abstract purpose: page 2, line 3

Objectives: To determine the association of sociodemographic characteristics and other related factors on attendance at diabetic eye screening in a large ethnically diverse urban population.

End of introduction section: page 4, line 32

We have examined attendance at diabetic eye screening to identify sociodemographic factors which are determinants of attendance in a large multi-ethnic population with high levels of deprivation.

2. You stated that your study design is retrospective cohort. However, I don't see any follow-up for patients attending the screening clinic. Can you explain further on the study variables/follow-up?

Thank you.

We have added more detail to the Setting subsection under the methods section to clarify this. The following change in Methods-setting section has been made:

Page 5, line 19

“Over the course of one year, every person eligible for Routine Digital Screening is offered multiple opportunities to attend. Eligibility for screening is defined by Public Health England.”

Thus, the cohort is defined as those eligible to be invited for Routine Digital Screening within the NHS during a defined time frame. Additionally, the diabetic eye screening programme is recommended annually and during the one year study period only 1 attendance visit can take place. In addition to the previous change, we have added also in our methods-setting section:

Page 5, line 31

“Any person attending any of the offered appointments over the course of one whole year was defined as ‘Attended’. Only those who failed to attend all appointments offered in the period were classified as ‘Did not attend’.”

We can now see that this should have been made clear for people not working in a screening programme.

3. The method is unclear. If it is a cohort study, I assume the follow-up the annual screening for each respondents. What did you do when patients come for repeated screening. Can you elaborate further on this? Thank you again for picking up this important point which we feel the above changes have addressed.

A study on repeated non-attendance would constitute a separate study with a different methodology/study period. Thank you for this future research suggestion.

4. If this is a cohort study, it should be relative risk not odd ratio with different time point. Given the dichotomous nature of our outcome measure over a restricted time frame of one year (i.e. “Attended” vs “Did not attend”), we have used logistic regression to analyse the odds of attendance, therefore, we report odds ratios.

Reviewer: 2

Mr. Md. Mehedi Hasan, University of Queensland

Comments to the Author:

This article is generally well-constructed and well-written. The findings and interpretations are interesting and seem to be grounded in the data. The results drawn from the data seems well-linked with the recommendations that may have role in further reducing health inequality on the topic. However, the authors may consider some minor issues as given below: Thank you for these positive comments.

1. “people with diabetes” may be replaced by “patients with diabetes” or “diabetic patients” throughout the paper Thank you. “People with diabetes” is the current preference of the major patient groups and organisations in the UK. Patients with diabetes might be clearer for physicians but probably not for other readers.

2. There should not be space between estimate and % in page 2 line 26 Amended.

3. There should be a space between 95% and CI in page 2 line 34-36 Also amended. Thank you

4. Outcome variable should be clearly stated in the text, should be in a separate paragraph. Thank you, we have included this in the first paragraph of the methods section.

Page 5, line 4

We performed a 12-month retrospective cohort study between 1st April 2017 to 31st March 2018. Our outcome measure is attendance at diabetic eye screening . Potential determinants of attendance include age, gender, self-defined ethnicity, area level deprivation, type of diabetes, duration of diabetes, visual acuity, years of NELDESP registration, distance to screening centre, and Public

Transport Accessibility. The study was registered and approved as an audit through the research governance process at the Homerton University Hospital NHS Foundation Trust.

5. Is there any cases for whom the residence was changed and result the change in their distance to screening center? If yes, how the distance was then adjusted? Thank you we agree that clarification is needed. We have added this on page 6 lines 15-18

Page 6, Line 11

We calculated distance to screening centre (in kilometres) as a straight line from the patient's postcode to the "screening site. For patients who attended, the postcode used was that known to the NELDESP on the day of attendance. For patients who failed to attend at any point within the study period, the postcode used was that known to the NELDESP on the date of the last offered appointment."

6. If possible, some variables such as medication intake for diabetes, controlled/uncontrolled diabetes should have an impact on attendance for screening, and hence should be considered The English NHS diabetic eye screening programme do not have routinely access to more patient data than is necessary to safely carry out the screening process. Medication and chemical pathology are held separately by patients' General Practitioners.

7. In page 6 line 24, which test was performed to examine trend of odds whether it is linear or not should be clearly mentioned Thank you. We have specified this in the mentioned page and line.

Page 6, line 27:

"A test for linear trend was performed if the odds ratios showed a reasonably linear pattern across categorical variables."

8. In page 7 line 36, not sure how the 6% rise was calculated. Would it be better to include a bit detail of the calculation? We have rephrased this sentence. For greater clarity we have made two additional changes:

a) The reference category for IMD is the most deprived quintile (1st IMD quintile) and this has been modified throughout.

b) Table 1 has been modified and the scores for continuous variables are shown in the table and correspond to what is discussed in the manuscript. Thank you.

(page7, line 33)

Change:

"Adjusted analyses showed that individuals living in the least deprived areas (5th IMD quintile) were most likely to attend for their screening appointments. Those in the 5th IMD quintile showed a 25% increase in the odds of attendance compared with people living in the most deprived areas (1st IMD quintile). Each IMD quintile increase (i.e. less deprivation) suggested a 6% rise in the odds of attendance (linear trend test p-value < 0.001)."

Change

Table 1, page 9 and 10.

Addition of rows with OR for continuous variables.

IMD reference group has been modified to most deprived quintile throughout.

9. The results of regression analysis were interpreted for univariable model in some places (for example, interpretation for visual acuity in page 7, line 53) and for multivariable model in other places (for example, interpretation for age in page 7, line 22). Should it be consistent? Thank you for pointing this out. This has been amended and we have reviewed that we are reporting consistently the odds ratios from the multivariable model. Additionally, we have added:

Page 7, line 18

“In the text we refer to the adjusted OR from the multivariable linear regression model unless otherwise stated.”

10. In the table, the results of regression analysis were presented when the independent variables were considered as factors, whereas some of these variables were not considered as factor variables when they were interpreted (e.g. age, distance etc). Should it be consistent with the estimates presented and those interpreted in the text? Rather than assuming linear associations for all potential determinants of attendance we first examine the patterns in the odds ratios by creating categories for each of the numerical data variables such as age for example. This allows one to examine possible deviations from linearity (see Table1, age 18 to 30 years category). In addition, we also provide the odds ratios per unit increase for continuous/numerical variables in table 1 (i.e. age per decade, duration of diabetes, distance to screening centre, each 5 years of registration, and visual acuity per 5 letters). This shows the size of the association if we treat these variables as continuous, and account for the possible loss of granularity with continuous variable categorization.

11. Some references are too old to use (such as ref 5, ref 6). Better to use updated one if possible. References have been updated throughout the manuscript.

VERSION 2 – REVIEW

REVIEWER	Hasan, Md. Mehedi University of Queensland, Institute for Social Science Research
REVIEW RETURNED	24-May-2021
GENERAL COMMENTS	Thank you for addressing the comments.